# Co-Occurrence of G6PD Deficiency and SCT among Pregnant Women Exposed to Infectious Diseases

**DOI:** 10.3390/jcm12155085

**Published:** 2023-08-02

**Authors:** Gideon Kofi Helegbe, Anthony Wemakor, Evans Paul Kwame Ameade, Nsoh Godwin Anabire, Frank Anaba, Jose M. Bautista, Bruno Gonzalez Zorn

**Affiliations:** 1Department of Biochemistry and Molecular Medicine, School of Medicine, University for Development Studies, Tamale P.O. Box TL 1883, Ghana; nanabire@uds.edu.gh; 2West African Centre for Cell Biology of Infectious Pathogens (WACCBIP), Department of Biochemistry, Cell, and Molecular Biology, University of Ghana, Legon, Accra P.O. Box LG 54, Ghana; 3Department of Nutritional Sciences, School of Allied Health Sciences, University for Development Studies, Tamale P.O. Box TL 1883, Ghana; 4Department of Pharmacognosy and Herbal Medicine, School of Pharmacy and Pharmaceutical Sciences, University for Development Studies, Tamale P.O. Box TL 1883, Ghana; 5Department of Biochemistry and Molecular Biology, Faculty of Biosciences, University for Development Studies, Nyankpala P.O. Box TL 1883, Ghana; 6Department of Biochemistry and Molecular Biology, Complutense University of Madrid, Ciudad Universitaria, 28040 Madrid, Spain; jmbau@ucm.es; 7Department of Animal Health, Complutense University of Madrid, Ciudad Universitaria, 28040 Madrid, Spain

**Keywords:** pregnant women, antenatal care, anemia, G6PD deficiency, sickle cell trait, co-occurrence, malaria, hepatitis B, syphilis, HIV

## Abstract

During pregnancy, women have an increased relative risk of exposure to infectious diseases. This study was designed to assess the prevalence of the co-occurrence of glucose-6-phosphate dehydrogenase deficiency (G6PDd) and sickle cell trait (SCT) and the impact on anemia outcomes among pregnant women exposed to frequent infectious diseases. Over a six-year period (March 2013 to October 2019), 8473 pregnant women attending antenatal clinics (ANCs) at major referral hospitals in Northern Ghana were recruited and diagnosed for common infectious diseases (malaria, syphilis, hepatitis B, and HIV), G6PDd, and SCT. The prevalence of all the infections and anemia did not differ between women with and without G6PDd (χ^2^ < 3.6, *p* > 0.05 for all comparisons). Regression analysis revealed a significantly higher proportion of SCT in pregnant women with G6PDd than those without G6PDd (AOR = 1.58; *p* < 0.011). The interaction between malaria and SCT was observed to be associated with anemia outcomes among the G6PDd women (F-statistic = 10.9, *p* < 0.001). Our findings show that anemia is a common condition among G6PDd women attending ANCs in northern Ghana, and its outcome is impacted by malaria and SCT. This warrants further studies to understand the impact of antimalarial treatment and the blood transfusion outcomes in G6PDd/SCT pregnant women.

## 1. Introduction

Human glucose-6-phosphate dehydrogenase (G6PD) is an X-linked housekeeping enzyme that catalyzes the first limiting step in the pentose phosphate pathway to generate reducing power in the form of NADPH, which is particularly necessary within the RBC cytoplasm to control its intrinsic oxidative stress [1,2]. The essentiality of G6PD to erythrocyte survival cannot be overemphasized since it is the exclusive producer of the coenzyme NADPH, which compensates for intracellular oxidative stress in the glutathione oxidoreduction cascade; so, in the absence of G6PD the cell dies [3]. G6PD deficiency (G6PDd) is a very common polymorphism, with about 400 million people carrying it in the world, and its prevalence is high in people of African, Asian, or Mediterranean descent [4]. There are more than 217 different polymorphic variants in its encoding gene described so far [5], and their carriers are predisposed to hemolytic crises by specific oxidative stress inducers, such as illness, some foods, or medication [2,6]. Although, hemolysis may be mild, moderate, or severe depending on the G6PD variant present [2,6,7], there is no specific treatment, other than avoiding known inducers [8]. While 32.5% of the G6PDd global prevalence has been reported for the sub-Saharan region [9], in Ghana the G6PDd prevalence ranges between 15 and 20% [10]. In particular, a study conducted by Owusu et al. of pregnant women in Ghana [7] reports that the G6PDd prevalence is differentiated into full defect (2.3%) and partial defect (17.0%) [11], accounting for the homozygous and heterozygous genetic dose of the ubiquitous G6PD A- variant in this region [12]. In connection with the landmark discovery that female erythrocytes are mosaic for G6PD due to inactivation of one of the X chromosomes [13], the high frequency of female anemia in the region [14] suggests the need for a close examination of the functional G6PD deficiency status of pregnant women. This fact goes beyond knowing only the presence of the genetic mutations that are present in homo- or heterozygosis, as functional deficiency may have implications not only for malaria but also for other endemic infections, and this has implications for targeted interventions aimed at pregnancy surveillance in this setting.

G6PDd distribution overlaps in the regions where malaria is predominant [4]. Thus, the use of antimalarials by G6PDd individuals in malaria-endemic regions can never be avoided and poses a threat to the health of vulnerable groups such as pregnant women and children. Most people with G6PDd are asymptomatic, but exposure to oxidant drugs, such as the antimalarial drugs quinine, primaquine, or sulfadoxine-pyrimethamine (SP), may induce hemolysis [8,15]. Even though infection (of viral or bacterial origin) also triggers hemolysis in G6PDd individuals [6], no data exist on the specific infections that may have this impact on G6PDd pregnant women.

Unlike G6PDd, sickle cell disease (SCD) is an autosomal recessive disorder, in which individuals inherit both abnormal beta-globin (HBB) alleles located on chromosome 11 (11p15.5). While about four million people are known to have SCD, forty-three million have sickle cell trait (SCT) [15], i.e., they inherit only one abnormal HBB allele. Close to 80% of SCD cases occur in sub-Saharan Africa [16] and other regions (India, the Middle East, and the Mediterranean) where malaria is or has been endemic, and the trait is known to confer protection against malaria disease [17,18]. The prevalence of SCT in Ghana has been estimated to be between 20% and 40% [19]. This has a great impact on clinical practice, ranging from management and treatment to blood transfusion. In particular, in pregnancy SCT has been associated with adverse maternal and neonatal outcomes, including pre-eclampsia, premature labor, bacteriuria in pregnancy, low birth weight, and fetal loss, as reviewed elsewhere [20]. In this regard, the data on the prevalence of SCT among pregnant women are useful in guiding policy on the diagnosis, management, treatment, and general health care given to pregnant women in antenatal clinics.

In summary, G6PDd and SCD/SCT are the most prevalent genetic polymorphisms in malaria-endemic areas and are characterized by high red blood cell turnover [2,21,22,23]. In Ghana, a malaria-endemic area, G6PDd and SCT co-inheritance is common, with a prevalence rate of 7% among blood donors [24]. Despite that, the co-occurrence of G6PDd (characterized by the RBCs’ inability to handle oxidative stress) with a hemoglobinopathy such as SCD or SCT can predispose individuals to enhanced oxidative stress, resulting in increased hemolysis and adverse events. To this effect, this study evaluated the prevalence of G6PDd/SCT co-occurrence and the associated risk factors, such as demographic factors, endemic infections, and anemia among pregnant women in northern Ghana.

## 2. Material and Methods

### 2.1. Sampling Population and Design

Study Area: The study was conducted in the ANC unit of the Bolgatanga Regional Hospital (Upper East Region), the Tamale Teaching Hospital (Northern Region), and the Wa Regional Hospital (Upper West), all in Northern Ghana (Figure 1). These three hospitals were chosen based on their location and because they are referral hospitals that provide health care services to the residents of Northern Ghana as well as the neighboring countries of Burkina Faso, Cote d’Ivoire, and Togo. The study was conducted between March 2013 and October 2019.

Study Design: The study design employed was a hospital-based cross-sectional type. Information on the gravidity, parity, age, and trimester of the visit of participants was obtained by means of a guided questionnaire via an interview. At the ANC, the pregnant women were bled twice; capillary samples were first obtained and used for rapid diagnostic testing to assess the seroprevalence of malaria, HBV, syphilis, and HIV; venous blood samples were subsequently collected at the laboratory and used for the quantification of hematological parameters (sickling, Hb, and the G6PD tests). The study subjects consisted of the first ANC attendees (pregnant women who gave their consent) in any trimester of pregnancy. The pregnant women were recruited as and when they appeared at the ANC, and the numbers obtained within the sampling period was used for the study. Women with documented chronic alcoholism and those with known chronic degenerative diseases with excluded from the study. The pregnant women diagnosed with the infections G6PDd, SCT, and/or anemia did not show any disease symptoms during the study period; despite that, they were referred by the midwife/nurse for appropriate treatment at the health facility.

### 2.2. Hematological Investigations

Estimation of hemoglobin (Hb): The Hb levels among the study subjects were estimated via full blood count (FBC). Five (5) mL of venous blood was collected into an EDTA tube. Using a sample roller, the venous blood was mixed with the anticoagulant (EDTA 2K, EDTA-3K, or EDTA-2Na) for a minute. The Sysmex XS-500i (Kobe, Japan) analyzed the blood sample while the Hb values were obtained from the FBC data. The anemia degree defined throughout this study follows the WHO criterion of Hb < 11 g/dL for pregnant women [25].

Sickling Test: The sickle cell slide test was used to determine the sickle cell status of the participants [26]. This test is easy to perform and only needs one reagent. However, it does not distinguish between sickle cell trait (SCT) and sickle cell disease (SCD) [26]. It is helpful when the HbS solubility filtration test cannot be conducted [26]. The test was conducted as previously described [26]. Briefly, a drop of well-mixed venous blood or capillary blood of the study subject was placed on a slide. In severely anemic participants (Hb < 7 g/dL), two drops of venous blood were used. An equal volume of a fresh reducing reagent (disodium disulfite) was added and mixed gently; then, it was covered with a cover glass and incubated at room temperature for an hour. Air bubbles were excluded. Positive and negative controls were set up alongside the sample and were taken through similar treatment. This preparation was then examined microscopically for sickle cells within 10–20 min after incubation. With the positive and negative controls as the standard, the test preparation was reported as “sickle cell test negative” or “sickle cell test positive”.

Glucose-6-phosphate Dehydrogenase (G6PD) Test: The methemoglobin reduction test [26] was used to diagnose G6PD deficiency. For this test, venous blood collected into EDTA anticoagulant tube was used. Because of the fact that the mature red cells’ low G6PD activity may be covered up by the reticulocytes’ higher levels of G6PD, blood samples with less plasma (sufficient plasma was removed until the packed cell volume (PCV) was about 0.40) were used for the participants diagnosed with severe anemia (Hb < 7 g/dL) [26]. The blood samples were kept at room temperature. Based on the protocol used, the G6PD status of each sample should be determined within eight (8) h of blood collection [26]; however, in this study the G6PD status of each sample was determined within 2 h of blood collection. The recruited pregnant women were classified as normal, or full defect (homozygous G6PD-deficient) after being screened for the G6PD activity via the methemoglobin reduction test [26]. Briefly, about 6 mL of fresh venous blood from each participant was collected into well-labelled EDTA tubes. With 3 small glass tubes labelled “Test”, “Normal”, and “Deficient”, 0.1 mL of freshly prepared sodium nitrite-glucose reagent was pipetted into each of the “Test” and “Deficient” glass tubes. Afterwards, 0.1 mL of methylene blue reagent was pipetted into the Test only, followed by 2 mL of the venous blood sample into each of the three glass tubes, and mixed well. All three (3) glass tubes containing the sample were then incubated at 35–37 °C for 90 min. Three other large glass tubes (15 mL capacity) were also labelled “Test”, “Normal”, and “Deficient” and 10 mL of distilled water was put into each. About 0.1 mL of the well-mixed sample from the three (3) small glass tubes (“Test”, “Normal”, and “Deficient”) was transferred to the large tubes, respectively. The contents in each large tube were examined for color change. A red solution was considered to have normal G6PD activity (no defect); a deep brown solution was considered to be full defect (homozygous G6PD-deficient); and a light/pale brown or a color that was midway between red and brown was considered to be partial defect (heterozygous G6PD-deficient). The G6PDd individuals in this study included both full defect (homozygous G6PD-deficient) and partial defect (heterozygous G6PD-deficient). We took precise measures to minimize the impact of the time taken for the test and the storage conditions on the results; so, all the samples were tested within an identical time range.

### 2.3. Rapid Diagnostic Testing

Malaria Testing: A rapid diagnostic test (RDT) (Standard Diagnostics, Inc., Suwon-si, Republic of Korea; Bioline Malaria Ag Pf/Pan FK 60) was used to diagnose malaria disease in the blood samples collected from the recruited study subjects (pregnant women). The RDT system detects *Plasmodium falciparum* histidine-rich protein 2 (PfHRP-2) and *Plasmodium* spp. lactate dehydrogenase (pLDH), guaranteeing detection under HRP-2 deletion. As *P. falciparum* is the main Plasmodium species in Ghana, the RDT kit chosen was specific for *P. falciparum*. *P. malariae* and *P. ovale* only account for 3.6 and 1.0%, respectively, of the diagnosed malaria in Ghana [27]. The reading and interpretations of each test strictly followed the manufacturer’s protocol. It should be noted that submicroscopic asymptomatic malaria may pass undetected by RDT, as recently described in Ghana [27,28].

Hepatitis B Testing: The HBV surface antigen RDT kit (Premier Co., Ltd., Nagpur, India, and Transnational Technologies Inc., Manchester, UK) was used for testing hepatitis in whole blood. This kit is specific for and sensitive to the identification of the hepatitis surface antigen, as specified by the manufacturers.

HIV Testing: A pretest counselling session was held for the pregnant woman before the HIV test was conducted. The HIV first response RDT kit (Premier Co., Ltd., Nagpur, India, and Transnational Technologies Inc., Manchester, UK) was used for initial testing on whole blood, while the OraQuick test kit (Premier Co., Ltd., Nagpur, India, and Transnational Technologies Inc., UK) was used for confirmatory testing on oral fluid.

Syphilis Testing: This was performed on whole blood using the TP (*Treponema pallidum*) kit (INNOVITA (Tangshan) Biological Technology Co., Ltd., Tangshan, China), following the manufacturer’s instructions. This kit is highly sensitive and specific in identifying *Treponema pallidum* species, as specified by the manufacturers.

For all the rapid diagnostic testing above, the testing, results reading, and interpretations were performed strictly according to the manufacturer’s instructions. A positive test showed two color bands at the control (C) and test (T) lines in the result window. A test was considered negative when only one color band was observed at the control (C) line. An invalid test showed no color band at either the “C” or the “T” line, or it showed a color band at the “T” line. Invalid tests were repeated with new test kits.

### 2.4. Statistical Analyses

The data were entered into a Microsoft Excel spreadsheet, which was imported unto SPSS (version 28) for analyses. Summary output tables of the percentage distribution were produced for the categorical variables. The association of G6PD status with background characteristics, infections, anemia, and sickling status was evaluated using the Pearson chi-square test and Fisher’s exact test. Multinomial logistic regression analysis was used to predict the associations of G6PDd with the infections (malaria, HBV, HIV, and syphilis) and the hematological indicators (SCT and anemia). Univariate analysis of variance was applied to evaluate the effect of interaction between infections and SCT status on the outcome of anemia among G6PDd women. Prior to the univariate analysis, Levene’s test was used to test for homogeneity of variance between the interacting factors, with *p* = 0.158. For all the tests of association, *p* < 0.05 was considered statistically significant.

## 3. Results

### 3.1. Background Characteristics of Study Subjects

The background characteristics of the pregnant women are summarized in Table 1. A total of 8473 pregnant women were recruited for the six-year study. Of these women, the majority (66.6%) were in the age bracket of 18–30 years old (Table 1). Close to half of the study subjects had had 1–2 previous pregnancies (49.0%) and 1 or 2 births (48.7%, Table 1).

### 3.2. Prevalence of G6PDd, Infections, Anemia, and SCT among Study Subjects

The prevalence of G6PDd, sickling positivity, and anemia was 8.6%, 7.2%, and 44.8%, respectively, for all the participants screened (Table 1). The prevalence of G6PDd and sickling co-defect was 0.008% (70/8473, Figure 2A). The seroprevalence of malaria, hepatitis B, HIV, and syphilis was 7.1%, 4.2%, 1.3%, and 0.6%, respectively (Table 1).

### 3.3. Association of G6PDd with Demographic and Clinical Factors

Age, gravidity, and parity had no significant association with G6PDd among the pregnant women (χ^2^ < 3.0; *p* > 0.05 for all comparisons, Table 2). Disease infections (malaria, HIV, HBV, and syphilis) and anemia status were not significantly associated with G6PDd (χ^2^ < 3.6; *p* > 0.05 for all comparisons, Table 2). However, sickling status was observed to be significantly associated with G6PDd status (χ^2^ = 15.7; *p* < 0.001, Table 2). Further analysis using multinomial logistic regression analysis revealed that pregnant women with G6PDd had a significantly higher proportion of co-occurrence with sickling positivity in comparison to the non-G6PDd women (AOR = 1.58; *p* < 0.011, Appendix A).

### 3.4. Co-Occurrence of Infections with Sickling Positivity among G6PDd Women and Its Association with Anemia

The distribution of the demographic and obstetric information among the G6PDd women is summarized in Appendix A. Anemia was diagnosed in 48.1% of the G6PDd women (Appendix A). The distribution of sickling positivity and infection among the G6PDd women is shown in Figure 2A. The percentages of G6PDd women with sickling positivity, malaria, hepatitis B, syphilis, and HIV were 9.6%, 5.2%, 3.9%, 0.3%, and 0.4%, respectively (Figure 2A), while in the normal G6PD women the percentages were quite comparable (Figure 2B). In the G6PDd women, the percentages of co-infections/defects of malaria*sickling, malaria*hepatitis B, malaria*syphilis, malaria*HIV, hepatitis B*sickling, HIV*sickling, and malaria*hepatitis B*sickling were 0.6%, 0.3%, 0.1%, 0.1%, 0.3%, 0.1%, and 0.3%, respectively (Figure 2A), while in the women with normal G6PD the percentages were quite comparable (Figure 2B). Additionally, the co-infections/defects of malaria*HIV*sickling (0.03%), HIV*syphilis (0.01%), syphilis*sickling (0.09%), hepatitis B*syphilis (0.04%), and hepatitis B*HIV (0.01%) were detected in women with normal G6PD (Figure 2B). Chi-square analysis showed that the proportions of the women with or without anemia were similar to those in the infection groups of the G6PDd women (χ^2^ = 18.47; *p* < 0.102, Figure 3A) and women with normal G6PD (χ^2^ = 22.83; *p* < 0.155, Figure 3B). Of the all analyzed infections, the interaction between malaria and sickling positivity was observed to be associated with anemia outcomes among the G6PDd women (F-statistic = 10.9, *p* < 0.001, Table 3).

## 4. Discussion

Anemia is associated with pregnancy [29,30], particularly from the first to the third trimester [31]; the WHO estimates that 40% of pregnant women are anemic worldwide [32]. Meanwhile, the condition is highly prevalent among women in their reproductive ages who reside in low- and middle-income countries (LMIC) [33]. Anemia is a great public health concern because it presents symptoms such as fatigue, dizziness, weakness, and shortness of breath, among others. Due to the instability of the G6PD A- variant enzyme [12], the individuals carrying it have increased susceptibility to hemolysis. As these G6PDd individuals often have lower Hb values, both situations together can lead to chronic anemic states that are not caused only by the presence of the G6PD A- variant [2,6]. This is in agreement with the increased percentages of the pregnant women carrying G6PDd who showed anemia in comparison the with normal G6PD women.

The impact of medication such as primaquine, quinine, and sulfadoxine pyrimethamine (SP), which trigger hemolysis in G6PDd individuals, is well documented [8]. Though infection is a known trigger for hemolysis in G6PDd, not many data exist on the type of infection that will trigger hemolysis in G6PDd individuals and pregnant women in general. In this study, infections such as malaria, hepatitis B, syphilis, and HIV and their co-infections were also investigated in G6PDd women. Our association analysis showed that the proportion of G6PDd women with or without anemia was similar across the different infection groups, which could be due to the fact that anemia was a common primary condition among these women. It was expected that infections such as malaria and HIV would be factors that triggered hemolysis due to their pathogenesis and disease outcome. While malaria parasite growth and development within RBCs and immune pressure cause hemolysis of RBCs, resulting in low Hb and anemia in malaria cases [34,35,36,37], low Hb in HIV is caused by bone marrow suppression [38], suppression of erythropoietin [39], and direct effects leading to RBC loss [40]. However, the interactive effect of a single infection of malaria, HIV, and hepatitis B was not associated with anemia outcome in G6PD-deficient women (Table 3). It can be hypothesized that these reported infections were mainly asymptomatic cases [28], thus viral load, parasitemia, or bacteremia would not be significantly high enough to initiate hemolysis in the G6PDd individuals. Additionally, malaria parasites require an optimum RBC redox status for their development, replication, and survival [41]. Although this precondition is notably limiting in G6PDd RBCs, it has been genetically balanced for a beneficial effect to protect against uncomplicated malaria in adult populations in sub-Saharan Africa [42]. It has been suggested that hepatitis B, on the other hand, plays a compensatory role in maintaining hemoglobin levels [30].

G6PDd and SCD are two common genetic conditions in malaria-endemic regions and are associated with protection against malaria [43]. Thus, the substantial degree of G6PDd/SCT co-inheritance should be considered as a public health concern for people of African descent and particularly for pregnant women [44]. In this study, we found a significant number of G6PDd women with sickle status (70/726); consequently, this also occurs the other way round, with sickle status women carrying G6PDd (70/613). Of the infections, the interactive effect between malaria and sickling positivity was observed to be associated with anemia outcomes in G6PDd women (Table 3). This interactive effect may suggest that asymptomatic malaria can influence the anemia outcome in pregnant women with a co-inheritance of G6PDd and SCT. Such a finding has implications for pregnancy follow-up, especially considering the meaningful number of pregnant women that require blood transfusion at delivery in Ghana [45].

Misdiagnosis is another potential challenge; there is a possibility of misdiagnosing G6PDd among sickling positive or SCD individuals. G6PD enzyme activity is elevated in reticulocytes, while it decreases as the RBCs age [2,6]. Considering that reticulocytosis is high in SCD [46], individuals with G6PDd/SCD co-inheritance are likely to show higher G6PD activity than those who do not have SCD [47]. In this regard, G6PDd/sickling positive individuals may have enzyme activity within the reference range [48,49]. This can have direct implications for those with the G6PDd/SCD co-occurrence, who are likely to be treated with antimalarial drugs (quinine, primaquine, and SP), which could trigger additional hemolysis with the attendant problems of hyperbilirubinemia and acute hemolytic anemia. This may additionally mask the original diagnoses of other anemias and thus generate false positive results for some of the anemic patients (reviewed by [6]).

Laboratory findings in patients with G6PDd and SCD show that reduced hemoglobin and hematocrit, a high level of reticulocytes, elevated lactate dehydrogenase (LDH), and increased unconjugated bilirubin are associated with increased hemolysis [50]. Monitoring these biomarkers is warranted when providing targeted treatment and management since, for example, high reticulocyte counts are associated with higher mortality in hemodialysis [51]. Thus, to prevent complications in pregnant women with the G6PDd/SCD co-inheritance that frequently presents reduced Hb levels, and particularly during acute hemolytic crises, blood transfusion should be the default immediate treatment [7,15]. Furthermore, considering hemolysis and anemia are risk factors for cerebral vasculopathy [6,51] and enhanced hemolysis in G6PDd patients, the close monitoring of Hb levels in G6PDd/SCD patients, and particularly among pregnant women, is suggested.

### Strengths and Limitations

The strength of the present study relies on the large dataset used, from which a convincing and sound conclusion was derived. However, some limitations are acknowledged, such as the lack of molecular genetic studies to identify the alleles associated with G6PDd/SCT co-occurrence that could have been misclassified in some cases of G6PDd, but with very a minor impact according to the large number of samples used in the study and given that it would only occur in cases concurrent with submicroscopic malaria that were undetectable by RDT. Thus, according to recent data [28], cases of submicroscopic malaria in pregnant women do not have large enough changes in hematological parameters for there to be an increase in G6PD activity in submicroscopically infected women that would substantially change the data in our statistical analysis. An additional limitation is our inability to homogenize data to investigate the treatment and pregnancy outcomes of the study population. Furthermore, the use of a cross-sectional study design may not be fully appropriate for studies of cause-and-effect relationships. Despite these limitations, we think that the study provides new insights into the co-occurrence of some common infections with frequent erythrocyte phenotypes among pregnant women in Ghana that can affect disease triggering and its treatment.

## Figures and Tables

**Figure 1 jcm-12-05085-f001:**
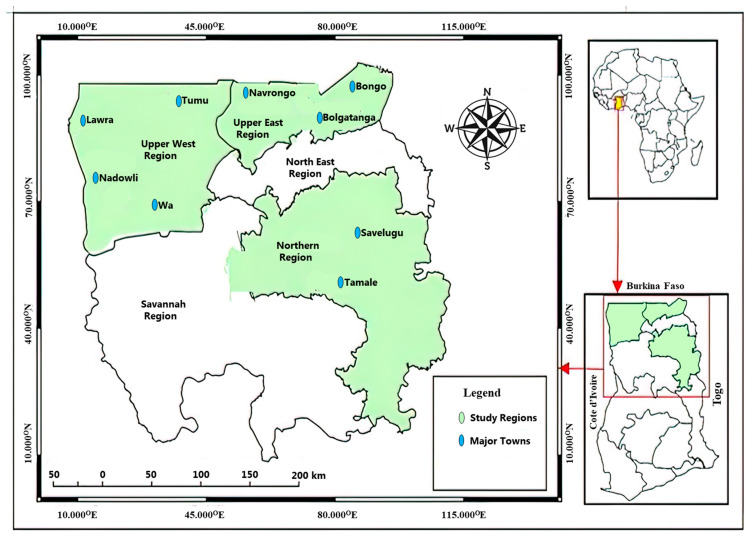
Map of study area.

**Figure 2 jcm-12-05085-f002:**
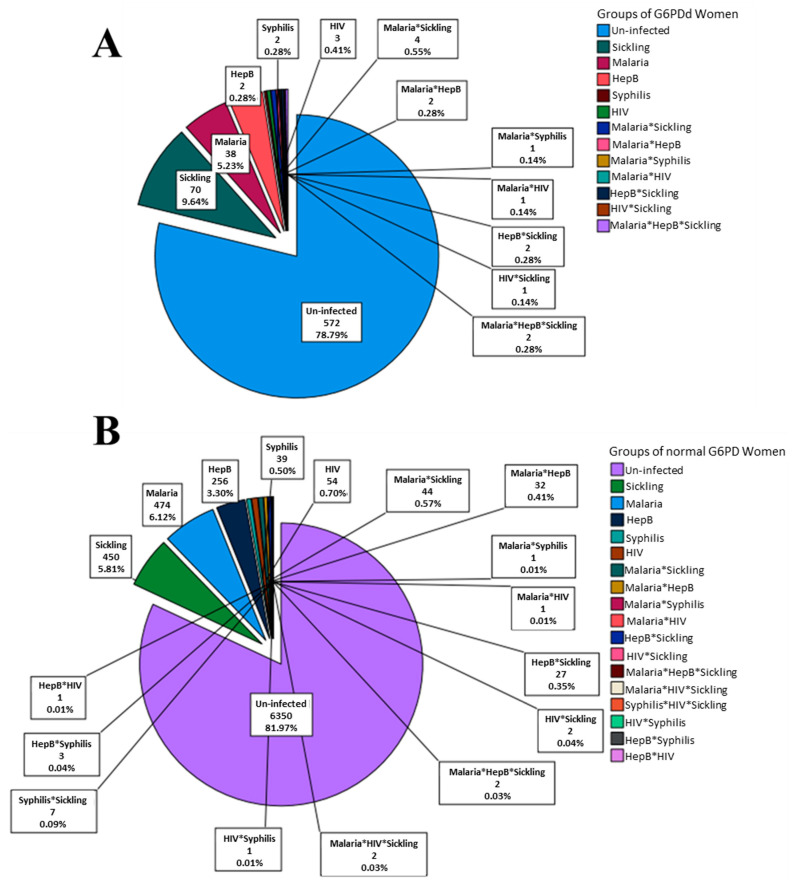
Distribution of co-occurrence of sickling positivity and infections among (**A**) women with G6PDd and (**B**) those with normal G6PD. * indicates co-infection/co-defect.

**Figure 3 jcm-12-05085-f003:**
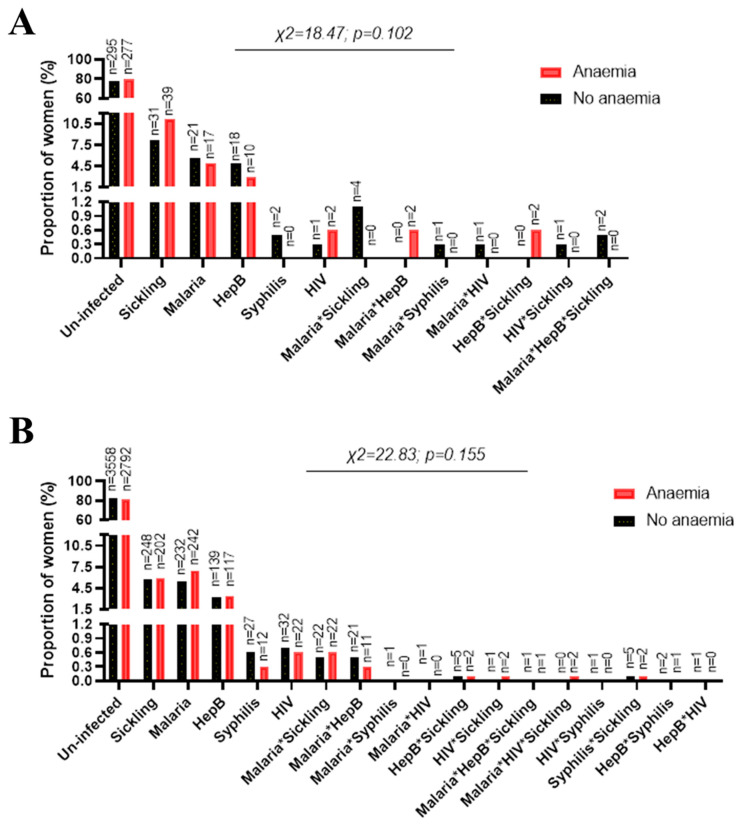
Association between infection groups and anemia among (**A**) G6PDd women and (**B**) those with normal G6PD. On the graphs, the chi-square value is indicated as χ^2^, and the *p*-values were obtained by Pearson chi-square test or Fisher’s exact test and considered significant at <0.05 (2-tailed). * indicates co-infection/co-defect.

**Table 1 jcm-12-05085-t001:** Age, pregnancy data, genetic status, anemia, and infections detected in the pregnant women screened.

Variable	Subgroup	Frequency	Percentage (%)
Demographic and obstetric information			
Age of respondent (years) (*n* = 8473)	<18	47	0.6
	18–25	2404	28.4
	26–30	3235	38.2
	31–40	2716	32.1
	41+	71	0.8
Gravidity (*n* = 8473)	0	9	0.1
	1–2	4156	49.0
	3–4	3020	35.6
	4+	1288	15.2
Parity (*n* = 8473)	0	2282	26.9
	1–2	4129	48.7
	3–4	1672	19.7
	4+	390	4.6
G6PDd, sickling, anemia, and infections statuses			
G6PDd (*n* = 8473)	Negative	7747	91.4
	Positive	726	8.6
Sickling (*n* = 8473)	Negative	7860	92.8
	Positive	613	7.2
Anemia (*n* = 8473)	Negative	4681	55.2
	Positive	3792	44.8
Malaria (*n* = 8473)	Negative	7869	92.9
	Positive	604	7.1
Syphilis (*n* = 8465)	Negative	8410	99.4
	Positive	55	0.6
HIV status (*n* = 5261)	Negative	5194	98.7
	Positive	67	1.3
Hepatitis B (*n* = 8473)	Negative	8117	95.8
	Positive	356	4.2

**Table 2 jcm-12-05085-t002:** Association of G6PD status with background characteristics, infections, anemia, and sickling status.

Variable	Subgroup	Total	G6PD-Non-Deficient	G6PD-Deficient	Chi-Square (χ^2^); *p*-Value
			Frequency (%)	Frequency (%)	
Demographic and obstetric information					
Age of respondent (*n* = 8473)	<18	47	41 (87.2%)	6 (12.8%)	χ^2^ = 2.9; *p* = 0.573
	18–25	2404	2201 (91.6%)	203 (8.4%)	
	26–30	3235	2953 (91.3%)	282 (8.7%)	
	31–40	2716	2484 (91.5%)	232 (8.5%)	
	40+	71	68 (95.8%)	3 (4.2%)	
Gravidity (*n* = 8473)	0	366	9 (2.5%)	357 (97.5%)	χ^2^ = 1.7; *p* = 0.642
	1–2	4050	3799 (93.8%)	251 (6.2%)	
	3–4	2887	2769 (95.9%)	118 (4.1%)	
	4+	1170	1170 (100.0%)	0 (0.0%)	
Parity (*n* = 8473)	0	2282	2082 (91.2%)	200 (8.8%)	χ^2^ = 1.2; *p* = 0.761
	1–2	4129	3772 (91.4%)	357 (8.6%)	
	3–4	1672	1539 (92.0%)	133 (8.0%)	
	4+	390	354 (90.8%)	36 (9.2%)	
Infections, anemia, and sickling statuses					
Malaria (*n* = 8473)	Negative	7869	7191 (91.4%)	678 (8.6%)	χ^2^ = 0.3; *p* = 0.571
	Positive	604	556 (92.1%)	48 (7.9%)	
Syphilis (*n* = 8465)	Negative	8410	7688 (91.4%)	722 (8.6%)	χ^2^ = 0.1; *p* = 0.729
	Positive	55	51 (92.7%)	4 (7.3%)	
HIV status (*n* = 5261)	Negative	5194	4792 (92.3%)	402 (7.7%)	χ^2^ = 0.0; *p* = 0.933
	Positive	67	62 (92.5%)	5 (7.5%)	
Hepatitis (*n* = 8473)	Negative	8117	7426 (91.5%)	691 (8.5%)	χ^2^ = 0.8; *p* = 0.384
	Positive	356	321 (90.2%)	35 (9.8%)	
Anemia (*n* = 8473)	Negative	4681	4304 (91.9%)	377 (8.1%)	χ^2^ = 3.5; *p* = 0.060
	Positive	3792	3443 (90.8%)	349 (9.2%)	
Sickling (*n* = 8473)	Negative	7860	7213 (91.8%)	647 (8.2%)	χ^2^ = 15.7; *p* < 0.001
	Positive	613	534 (87.1%)	79 (12.9%)	

Proportions were compared by chi-square analysis; χ^2^ indicates the chi-square value; *p*-value was obtained by Pearson chi-square analysis and considered significant at < 0.05 (2-tailed).

**Table 3 jcm-12-05085-t003:** Tests of between-subject effects on the outcome of anemia among G6PDd women.

Factor	Mean Squares	F-Statistic	*p*-Value
Sickling	0.420	1.696	0.193
Malaria	0.448	1.808	0.179
Hepatitis B	0.424	1.713	0.191
Syphilis	0.727	2.935	0.087
HIV	0.400	1.614	0.204
Malaria * Sickling	2.702	10.910	0.001
Hepatitis B * Malaria	0.034	0.139	0.709
Malaria * Syphilis	0.001	0.004	0.952
Malaria * HIV	0.291	1.177	0.278
Syphilis * Sickling	0.004	0.017	0.895
Hepatitis B * Sickling	4.929 × 10^−3^	0.000	0.989
HIV * Sickling	0.414	1.673	0.196
Malaria * Hepatitis B * Sickling	0.712	2.875	0.090

Dependent variable: anemia. The F-statistic tests the effects of the groups on anemia outcomes. The test is based on the nearly independent pairwise comparisons among the estimated marginal means. * indicates co-infection/co-defect.

## Data Availability

Raw data used and/or analyzed during the current study are available from the corresponding author on reasonable request.

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
