# Peer review of "Co-Occurrence of G6PD Deficiency and SCT among Pregnant Women Exposed to Infectious Diseases"

_jcm, 2023, doi:10.3390/jcm12155085_

Round 1

Reviewer 1 Report

The data are strong and well presented, however the results are not relevant. The study is descriptive and not suitable for the journal scope due to the lack of information about mechanistics or treatments.

Author Response

REVIEWER 1: Comments and Suggestions for Authors

The data are strong and well presented, however the results are not relevant. The study is descriptive and not suitable for the journal scope due to the lack of information about mechanistics or treatments.

Response: Our manuscript provides new insights into the co-occurrence of some common infections with frequent erythrocyte phenotypes among pregnant women in Ghana that can affect disease triggering and its treatment. The findings has implications on the health care delivery given to pregnant women on routine antenatal clinics in malaria endemic areas, particularly Northern Ghana. Considering, the findings from study can have implications on treatment options of expectant mothers seeking prenatal services, we disagree with the reviewer that our study does not provide any information of treatments and that it is not suitable to be considered for publication by journal

Reviewer 2 Report

The authors have represented a well-planned and executed study about G6PD deficiency and SCT co-occur in pregnant women exposed to infectious diseases. This calls for more research into the effects of antimalarial treatment and blood transfusion outcomes in G6PDd/SCT pregnant women.

Author Response

REVIEWER 2: Comments and Suggestions for Authors

The authors have represented a well-planned and executed study about G6PD deficiency and SCT co-occur in pregnant women exposed to infectious diseases. This calls for more research into the effects of antimalarial treatment and blood transfusion outcomes in G6PDd/SCT pregnant women.

Response: We thank the reviewer for highlighting the strengths of our study. Indeed, our findings prompts the need to evaluate the antimalarial treatment and blood transfusion outcomes in pregnant women diagnosed with G6PDd/SCT.

Reviewer 3 Report

This manuscript shows results of a cross-sectional study, that included as participants, a large number of pregnant women, that visited three-reference hospitals in Ghana. The authors evaluated 8,473 pregnant women. The main goal was to evaluate the prevalence of G6PDd/SCT co-occurrence and associated risk factors such as demographic factors, endemic infections and anemia among pregnant women in Northern Ghana.

The manuscript has interesting results, but it needs to be improved in order to be published.

Considerations:

Line: 104: Map of the study area: The map is not good. The resolution is too low and the names are blurred. Add the name of the surroundings countries

Line 111: Information about study design is too short and despite the large number of participants, it does not allow evaluating the external validity of the study.

For example, it is necessary to add the criteria of sample inclusion and exclusion. When were the women approached to participate in the study? How was calculated the size sample? Authors worked with the populational sample or with the universe of pregnant woman?

On the other hand, authors said that “The pregnant women diagnosed of the infections, genetic polymorphisms (G6PDd and SCT)… “. Strictly speaking, the authors did not assess genetic polymorphisms. They evaluated the prevalence of G6PDd (using the Methaemoglobin reduction test) and SCD (measured by the sickle cell slide test). No genetic polymorphisms experiments were carried out.

Line 141: PCV: What is that? Pressure central venous? The first time that an acronym be made, it may be described by extensive.

Line 219: Authors said: …“Age, gravidity and parity had no significant association with G6PDd among the 219 pregnant women (χ2<3.0; p<0.05 for all comparisons)”. Authors have a mistake with the p-value. a value of p<0.05 shows a statistically significant result. In this sentence, they said that a p<0.05 had no significant association with G6PDd. Please clarify and to add correct interpretation.

Line 221: Authors said: … “were not significantly associated with G6PDd 221 (χ2<3.6; p<0.05 for all comparisons, Table 2)”. same comment as in the previous sentence. P<0.05 is a statistically significant result. Please correct.

Line 234-252: This paragraph is too long and repeat information. As these results are shown in Figure 2A and B, Figure 3 and Table 3, I think that is better to do a short review of the results here. It is not necessary to show all data, only the most important findings.

Line 265: In general, the discussion needs to be more in-depth. The data should be better used in light of the world literature. For example, despite being important diagnostic tools, especially in resource-poor areas, RDTs have a number of limitations. Most RDTs used for malaria, need a minimum parasitemia of 100 parasites per μL to detect infection. Thus, the patient may have a plasmodial infection with less than 100 parasites/μL and the test be negative (false negative) even though the patient is infected. The authors nowhere discuss this limitation to their results. The same reasoning can be done for the other RDT performed.

Line 307: Authors said: “Taken together, these results may suggest that asymptomatic malaria can exacerbate anemia in pregnant women with co-inheritance of G6PDd and SCT”. However the study design and experimental approach do not allow observing this evidence. In which part of the results, do the authors show these findings of asymptomatic infection?

Line 314: Authors said: “Considering reticulocytosis is high in SCD…”. Did the authors measure reticulocytoses? If not, why not?

In general, the study was conducted in three referral hospitals in northern Ghana. There are no analyzes on differences that may have existed, according to the recruiting hospital. Is the epidemiological context for the prevalence of the examined diseases the same in all hospitals? Did authors find any difference in their results when each hospital is evaluated? These data can provide new interesting results.

Author Response

REVIEWER 3: Comments and Suggestions for Authors:

This manuscript shows results of a cross-sectional study, that included as participants, a large number of pregnant women, that visited three-reference hospitals in Ghana. The authors evaluated 8,473 pregnant women. The main goal was to evaluate the prevalence of G6PDd/SCT co-occurrence and associated risk factors such as demographic factors, endemic infections and anemia among pregnant women in Northern Ghana.

The manuscript has interesting results, but it needs to be improved in order to be published.

Response: As much as possible we have improved the manuscript by critically addressing the reviewer’s comments

Considerations:

Line: 104: Map of the study area: The map is not good. The resolution is too low and the names are blurred. Add the name of the surroundings countries

Response: This has been fixed. The names of the surrounding countries have been included as well (line 103)

Line 111: Information about study design is too short and despite the large number of participants, it does not allow evaluating the external validity of the study.

For example, it is necessary to add the criteria of sample inclusion and exclusion. When were the women approached to participate in the study? How was calculated the size sample? Authors worked with the populational sample or with the universe of pregnant woman?

Response: As suggested by the reviewer, we have included the criteria for recruitment of the study participants. Also, we also addressed how the sample was obtained (see lines 112 – 115).

On the other hand, authors said that “The pregnant women diagnosed of the infections, genetic polymorphisms (G6PDd and SCT)… “. Strictly speaking, the authors did not assess genetic polymorphisms. They evaluated the prevalence of G6PDd (using the Methaemoglobin reduction test) and SCD (measured by the sickle cell slide test). No genetic polymorphisms experiments were carried out.

Response: We appreciate the reviewer's insightful comment. We concur with the reviewer and have removed ‘genetic polymorphisms’ from the text.

Line 141: PCV: What is that? Pressure central venous? The first time that an acronym be made, it may be described by extensive.

Response: PCV stands for packed cell volume, which has already been defined on line 143.

Line 219: Authors said: …“Age, gravidity and parity had no significant association with G6PDd among the 219 pregnant women (χ2<3.0; p<0.05 for all comparisons)”. Authors have a mistake with the p-value. a value of p<0.05 shows a statistically significant result. In this sentence, they said that a p<0.05 had no significant association with G6PDd. Please clarify and to add correct interpretation.

Response: The error has been fixed, and the value is now p>0.05 (line 225).

Line 221: Authors said: … “were not significantly associated with G6PDd 221 (χ2<3.6; p<0.05 for all comparisons, Table 2)”. same comment as in the previous sentence. P<0.05 is a statistically significant result. Please correct.

Response: The error has been fixed, and the value is now p>0.05 (line 227).

Line 234-252: This paragraph is too long and repeat information. As these results are shown in Figure 2A and B, Figure 3 and Table 3, I think that is better to do a short review of the results here. It is not necessary to show all data, only the most important findings.

Response: This section of the result presentation is the main thrust of our investigation on the co-occurrence of G6PDd/SCT in women with the different infections, and as such we have duly presented the data to guide an informed perusal by readers. We have deleted some portions and kept a majority of the text as they highlight important findings (lines 239 – 256).

Line 265: In general, the discussion needs to be more in-depth. The data should be better used in light of the world literature. For example, despite being important diagnostic tools, especially in resource-poor areas, RDTs have a number of limitations. Most RDTs used for malaria, need a minimum parasitemia of 100 parasites per μL to detect infection. Thus, the patient may have a plasmodial infection with less than 100 parasites/μL and the test be negative (false negative) even though the patient is infected. The authors nowhere discuss this limitation to their results. The same reasoning can be done for the other RDT performed.

Response: in our discussion we have highlighted the limitations of RDTs as they may not be able to pick up submicroscopic infections (lines 340 – 348). That notwithstanding we continue to make a case on why the missed infections might not have an overall impact on our results. Since cases of submicroscopic malaria in pregnant women do not cause significant changes in haematological parameters as demonstrated in our previous work [ref 28], it is unlikely that an increase in G6PD activity in submicroscopically infected women will significantly alter the data in our statistical analysis.

Line 307: Authors said: “Taken together, these results may suggest that asymptomatic malaria can exacerbate anemia in pregnant women with co-inheritance of G6PDd and SCT”. However the study design and experimental approach do not allow observing this evidence. In which part of the results, do the authors show these findings of asymptomatic infection?

Response: The statement has been deleted. We have replaced the statement by cautiously stating that asymptomatic malaria may influence anaemia outcomes based on our finding that the interactive effect between malaria and sickling positivity was associated with anaemia outcomes in G6PDd women (Table 3) (see on lines 311 – 313)

Line 314: Authors said: “Considering reticulocytosis is high in SCD…”. Did the authors measure reticulocytoses? If not, why not?

Response: Reticulocytosis was not measured. A suitable citation for the made assertion has been added [ref 46] (line 318). SCT and SCD were not distinguished in our study since genotyping was not done. The comment was made to advance discussion on the relative levels of G6PD activity in situations of G6PDd/SCD co-inheritance and in cases of SCD alone (lines 318–320).

In general, the study was conducted in three referral hospitals in northern Ghana. There are no analyzes on differences that may have existed, according to the recruiting hospital. Is the epidemiological context for the prevalence of the examined diseases the same in all hospitals? Did authors find any difference in their results when each hospital is evaluated? These data can provide new interesting results.

Response: The epidemiological context is similar across various part of northern Ghana. The prevalence of G6PDd, SCT, and infections for this study was comparable across the three referral hospitals. Due to the dearth of data from the northern part of Ghana, the study was especially interested in the tale of the northern part of Ghana as a whole.

Reviewer 4 Report

Authors study the co-occurrence of the G6PD deficiency, sickle cell trait, anemia and various infections among pregnant women in Ghana. Number of participants is pleasantly high. This study is important for physicians, attracting the attention for the therapy by antimalarial drugs capable to elicit the additional hemolisis (anemia) and providing the ideas for targeted antimalarial treatment of G6PDd and SCD pregnant woman. Additionally, this study provides new insights into infection, anemia, G6PDd and SCD co-occurrence in Ghana.

Minor critics:

1.       One of the main result in this paper is anemia frequency, maybe it's worth including anemia in the title?

2.       Authors’ affiliation: for the authors from Tamale, the country, Ghana, is missing.

3.       Abstract: define ANC acronym when first time used.

4.       Introduction, line 45: The citation [4] for G6PDd world carriers is from 2009. Hope, more recent data are available. The same for SCT, line 74, the citation [16] is from 2010.

5.       Figure 1. In actual figure size/resolution, the name of the cities in the main map are poorly readable. Please, fix it.

6.       Methods, line 137, G6PD test chapter. Please, indicate, in which step of erythrocyte washing the white blood cells were removed to insure G6PD detection only in erythrocytes.

7.       Methods, line 137, G6PD test chapter. The test for G6PD deficiency is performed (as follows from text) by eye distinguish between red, light brown and deep brown colour of the tube. This “ancient” method for colour detection is used in the literature until today, thus roughly could be accepted in this paper. In the future the authors could introduce the spectrometric quantitative evaluation.

8.       Methods, line 137, G6PD test chapter. Both G6PD homozygotes and heterozygotes are included in statistics as 8473 G6PDd persons? Please, indicate it clearly.

9.       Methods, line 168, please, spell out the full term of HRP and LDH acronyms.

10.   Methods, lines 174, 178 and 183. Which type of primary human samples (blood?) were used for HBV, HIV and Treponema identification?

11.   Results, line 236. The Table S2 reports the important information about anemia percentage in G6PDd, probably worse to use this Table in the main text, non-supplemental?

12.   Discussion, line 266. The initial phrase “Anemia is associated with pregnancy…” is overstatement, better to add the clarification “…in some conditions” or similar.

13.   References, line 423, the [16] authors are listed as: “Tinley, K.E., A.M. Loughlin, A. Jepson, E.D. Barnett, and hygiene”, is it typo?

Author Response

REVIEWER 4: Comments and Suggestions for Authors

Authors study the co-occurrence of the G6PD deficiency, sickle cell trait, anemia and various infections among pregnant women in Ghana. Number of participants is pleasantly high. This study is important for physicians, attracting the attention for the therapy by antimalarial drugs capable to elicit the additional hemolisis (anemia) and providing the ideas for targeted antimalarial treatment of G6PDd and SCD pregnant woman. Additionally, this study provides new insights into infection, anemia, G6PDd and SCD co-occurrence in Ghana.

Minor critics:

  1. One of the main result in this paper is anemia frequency, maybe it's worth including anemia in the title?

Response: We appreciate the reviewer's efforts to improve our manuscript. Although anaemia is one of the study's primary findings, we still want to keep the title because it captures the study's main focus.

  1. Authors’ affiliation: for the authors from Tamale, the country, Ghana, is missing.

Response: Ghana has been included to the affiliations (see lines 6 – 10)

  1. Abstract: define ANC acronym when first time used.

Response: This correction has been effected (see line 23)

  1. Introduction, line 45: The citation [4] for G6PDd world carriers is from 2009. Hope, more recent data are available. The same for SCT, line 74, the citation [16] is from 2010.

Response: As much as possible we have cited the most updated literature per our search.

  1. Figure 1. In actual figure size/resolution, the name of the cities in the main map are poorly readable. Please, fix it.

Response: This has been fixed (line 103)

  1. Methods, line 137, G6PD test chapter. Please, indicate, in which step of erythrocyte washing the white blood cells were removed to insure G6PD detection only in erythrocytes.

Response: The method as described in the manuscript (following the protocol described in ref [26]) does not involve erythrocyte washing. However, in order to do away with white blood cells, we used pack blood (as described in the manuscript we used increased PCV of 0.4 (40%)) which corresponds to an increased red cell count (see lines 141 – 144).

  1. Methods, line 137, G6PD test chapter. The test for G6PD deficiency is performed (as follows from text) by eye distinguish between red, light brown and deep brown colour of the tube. This “ancient” method for colour detection is used in the literature until today, thus roughly could be accepted in this paper. In the future the authors could introduce the spectrometric quantitative evaluation.

Response: This method was employed in the current investigation because it is the method most frequently used to diagnose G6PDd in the study areas. We acknowledge that there is a chance that human error contributed to the results, but there were enough controls in place to reduce this risk, and three independent laboratory technicians all agreed on the results. Additionally, to reduce this risk of misclassification, the study classified G6PDd individuals to include both full defect (homozygous G6PD deficient) and partial defect (heterozygous G6PD deficient) (see lines 164 – 165). We agree with the reviewer’s recommendation to apply spectrometric quantitative evaluation in subsequent studies in our study areas.

  1. Methods, line 137, G6PD test chapter. Both G6PD homozygotes and heterozygotes are included in statistics as 8473 G6PDd persons? Please, indicate it clearly.

Response: This has been clearly stated in the methodology section (lines 152 – 163). That notwithstanding we have insert the sentence ‘’G6PDd individual in this study include both full defect (homozygous G6PD deficient) and partial defect (heterozygous G6PD deficient)’’ in lines 164 – 165 to bring more clarity.

  1. Methods, line 168, please, spell out the full term of HRP and LDH acronyms.

Response: These acronyms have been spelt out as suggested by the reviewer (see lines 172 – 173)

  1. Methods, lines 174, 178 and 183. Which type of primary human samples (blood?) were used for HBV, HIV and Treponema identification?

Response: Whole blood was used while oral fluid was used for confirmatory detection of HIV (See lines 180, 185 – 188)

  1. Results, line 236. The Table S2 reports the important information about anemia percentage in G6PDd, probably worse to use this Table in the main text, non-supplemental?

Response: We thank the reviewer for this critical comment. Once the prevalence of anaemia has been stated in pros in the main text, we would like to keep the table as supplementary data.

  1. Discussion, line 266. The initial phrase “Anemia is associated with pregnancy…” is overstatement, better to add the clarification “…in some conditions” or similar.

Response: This statement mainly talks about an association but not causality, and we have supported the statement with relevant references.

  1. References, line 423, the [16] authors are listed as: “Tinley, K.E., A.M. Loughlin, A. Jepson, E.D. Barnett, and hygiene”, is it typo?

Response: This was a mistake, thus all references have been formatted and any duplicate references have also been deleted.

Round 2

Reviewer 1 Report

Im asking to change reviewer